# Antiplatelet Activity of Coumarins: In Vitro Assays on COX-1

**DOI:** 10.3390/molecules26103036

**Published:** 2021-05-19

**Authors:** Cristina Zaragozá, Francisco Zaragozá, Irene Gayo-Abeleira, Lucinda Villaescusa

**Affiliations:** Pharmacology Unit, Biomedical Sciences Department, University of Alcalá, Alcalá de Henares, 28871 Madrid, Spain; francisco.zaragoza@uah.es (F.Z.); irene.gayo@uah.es (I.G.-A.); lucinda.villaescusa@uah.es (L.V.)

**Keywords:** coumarin, esculin, esculetin, antiplatelet activity, impedance aggregometry, COX, polyphenols

## Abstract

Atherosclerotic cardiovascular disease is the leading cause of death in developed countries. Therefore, there is an increasing interest in developing new potent and safe antiplatelet agents. Coumarins are a family of polyphenolic compounds with several pharmacological activities, including platelet aggregation inhibition. However, their antiplatelet mechanism of action needs to be further elucidated. The aim of this study is to provide insight into the biochemical mechanisms involved in this activity, as well as to establish a structure–activity relationship for these compounds. With this purpose, the antiplatelet aggregation activities of coumarin, esculetin and esculin were determined in vitro in human whole blood and platelet-rich plasma, to set the potential interference with the arachidonic acid cascade. Here, the platelet COX activity was evaluated from 0.75 mM to 6.5 mM concentration by measuring the levels of metabolites derived from its activity (MDA and TXB_2_), together with colorimetric assays performed with the pure recombinant enzyme. Our results evidenced that the coumarin aglycones present the greatest antiplatelet activity at 5 mM and 6.5 mM on aggregometry experiments and inhibiting MDA levels.

## 1. Introduction

Platelets present a wide variety of functions in the blood circulation, with a key role in the development of the atherosclerotic process and the subsequent physiopathology of the cardiovascular disease [1]. Once attached to the vascular endothelium and activated, the platelets release a broad range of molecules such as chemokines, proinflammatory agents and different substances able to modulate a biological response that will promote the interaction among platelets, endothelial cells and leukocytes [2]. These cell interactions trigger a local inflammatory response, which is mainly responsible for the atherosclerotic process [3]. Platelet adhesion to the luminal vascular surface occurs after exposure of the endothelium caused by a lesion or detachment of an atherosclerotic plaque. Platelet aggregation represents the initial stage in the formation of a blood clot that can lead, to a greater or lesser extent, to the vascular occlusion and eventually result in thromboembolic disease such as stroke or myocardial infarction [4].

An irreversible stage of platelet aggregation is mainly induced by the secretion of substances from the platelet granules content. This event has also been observed in vitro as a response to the addition of high concentrations of agonists. The process includes the formation of metabolites mostly derived from arachidonic acid (cyclic endoperoxides and TXA_2_) and the secretion of the content from lysosomes and dense and α-granules in platelets [5]. Coumarins (2*H*-1-benzopyran-2-ones), the lactones of the 2-hydroxy-Z-cinnamic acids, are phenolic compounds with complex structures that differ substantially across the family [6] and are extensively distributed in the plant kingdom, especially in the families *Apiaceae, Asteraceae* and *Rutaceae* [7]. Naturally occurring coumarins, even though all of them contain the coumarin moiety, are structurally different and can be classified according to their chemical structure in the following groups: simple coumarins, furanocoumarins, dihydro-furanocoumarins, phenylcoumarins, pyranocoumarins and dicoumarins [8].

Simple coumarins are usually substituted at position 7 (C-7) with a hydroxyl but can be also hydroxylated at positions 6 and 8. These hydroxyl groups can be sometimes methylated or substituted with sugar molecules, in which case they are referred to as glycosylated or heterosidic coumarins [9]. The presence of the different substituents in the main structure largely influences the biological activity of the resulting compound [10].

Coumarin (2*H*-1-benzopyran-2-one) (Figure 1) has been under research due to its interesting and wide-ranging bioactivities, inclusive of anti-inflammatory [11,12], antioxidant [7], antimicrobial [13,14,15], antiproliferative [16,17] and anticoagulant properties [18]. The vitamin K antagonists in clinical use are structurally derived from 4-hydroxycoumarin and share a common mechanism of action in that they noncompetitively inhibit the vitamin K epoxide reductase complex, which is essential in the recycling of vitamin K in the liver. As vitamin K serves as a cofactor in the activation of clotting factors II, VII, IX and X, the inhibition of its recycling results in strong anticoagulation activity [19]. 

Esculin (7-hydroxy-6-[(2 S, 3 R, 4 S, 5 S, 6 R)-3,4,5-trihydroxy-6-(hydroxymethyl) oxane-2-yl] oxychromen-2-one) (Figure 1) is a coumarin derivative found in *Aesculus hippocastanum* L. (horse chestnut) [20] that has demonstrated promising anti-inflammatory, antioxidant and free radical scavenging properties. This compound was effective in diminishing the elevated blood creatinine levels in diabetic mice, which ameliorated diabetes-induced renal dysfunction through a reduction on the activation levels of caspase-3 in the mice kidney [21]. Likewise, esculin showed a protective effect against lipid metabolism disorders in diabetic rats in a dose-dependent manner. The authors of this study proposed that the possible mechanism might be associated with the inhibition of AGE (advanced glycation end products) formation [22].

Conversely, esculetin (6,7-dihydroxychromen-2-one) (Figure 1) is the aglycone of the heteroside esculin. This compound has been thoroughly investigated because of its anti-inflammatory activity, which is conducted through several mechanisms that include the inhibition of ICAM-1 release, the decrease of NO and PGE_2_ levels in synovial fluid, myocardial protection or the inhibition of proinflammatory cytokines during the interaction between adipocytes and macrophages [23]. Some evidence for the potential of this aglycone to decrease oxidative stress has also been demonstrated [24], together with the presence of antidiabetic [25], antibacterial [26] and antitumor activities [27].

Despite the significant number of studies based on these types of chemical compounds and the diverse biological activities described for coumarin, esculetin and esculin, the mechanisms of action remain partially unknown. This research work focuses on demonstrating the antiplatelet activity of these coumarins and shedding some light on their mechanism of action. Due to the important role of the cyclooxygenase (COX) enzyme in platelet aggregation, it has been hypothesized that the potential interaction with COX is a possible mechanism through which coumarins could exert their antiplatelet function.

## 2. Results

### 2.1. Antiaggregant Effect of Coumarins by Impedance Platelet Aggregometry 

The percentage of platelet aggregation in whole blood (WB) and platelet-rich plasma (PRP) after activation by adenosine diphosphate (ADP) or arachidonic acid (AA) was calculated. Maximal aggregation (100%) was considered when ADP or AA were used in absence of any other compound. All the assayed phenolic compounds were tested at different concentrations: 0.75, 1.5, 3.0, 5.0, 6.5 mM. These concentrations are similar to the daily dose clinically used for the flavonoid diosmin (Daflon^®^ 500 mg) [28]. The percentages of aggregation for coumarin, esculin and esculetin are shown in Figure 2. 

In general, the antiplatelet effect of the screened phenolic compounds was observed at concentrations equal to or higher than 3 mM and was more potent in samples subjected to AA-induced platelet aggregation (Figure 2A,C) than in those subjected to ADP-induced platelet aggregation (Figure 2B,D), reaching on WB AA-induced experiments a IC50 for coumarin and esculetin of 2.45 mM and 3.07 mM respectively, and 5.12 mM and 5.82 mM on ADP experiments. The IC50 on PRP AA-induced samples were 1.12 mM for coumarin and 2.48 mM and 5.08 mM and 5.97 mM for ADP-induced assays.

The effects of coumarin and esculetin as antiaggregant agents were especially relevant in all the experiments performed. The complete inhibition of the platelet aggregation was achieved in AA-induced activated PRP samples after addition of 1.5 mM of coumarin (Figure 2C). 

### 2.2. Platelet MDA Levels and COX Activity

Considering that the extent of inhibition was greater in the impedance aggregometry assays when AA was used as activator (Figure 2A,C) [29], the MDA levels were quantified in AA-induced activated PRP samples. Maximal activity of COX (100%) was set as that obtained for AA-treated samples in the absence of phenolic compound. Indomethacin was used as a positive control at the doses shown below (Figure 3).

The inhibition of COX activity was confirmed in presence of indomethacin (Figure 3). Coumarin and esculetin showed their ability to inhibit COX-1 activity, while esculin did not affect it significantly (Figure 3). The IC50 for coumarin and esculetin were 5.93 mM and 2.76 mM for coumarin and esculetin, respectively. 

### 2.3. COX-1 Inhibitory Assay

With the aim to determine the direct effect of the compounds here investigated on the COX-1 activity, analyses with the pure human recombinant enzyme (h-COX-1) were performed. Results were expressed as the percentage of COX-1 activity in the presence of indomethacin (as a positive control) or after incubation with the different tested molecules. 

As it can be observed in Figure 3, indomethacin diminished almost completely the activity of the recombinant COX-1 enzyme. Coumarin produced a decrease of 49% in h-COX-1, while esculetin barely reached a drop of 42%. By contrast, esculin demonstrated the greatest COX-1 inhibitory effect, with a 74% of enzyme inhibition rate (Figure 4) and IC50 of 4.49 mM.

### 2.4. TXB_2_ Levels as COX-1 Activity Indicator

TXB_2_ quantification was conducted in WB and used as an indicator of the COX-1 activity since it is assumed that the administration of a COX inhibitor will decrease the levels of TXB_2_. The effect of indomethacin as a positive control was analysed at different concentrations. Meanwhile, the calcium ionophore (CI) (25 mM) was employed as a platelet aggregation inducer to evaluate the potential effect of the assayed phenolic compounds.

The results demonstrated the inhibition of the TXB_2_ production by indomethacin at all the concentrations tested. On the contrary, any of the phenolic compounds investigated exerted any effect on the levels of this metabolite (Figure 5).

## 3. Discussion

In this research study, the potency of the aglycones coumarin and esculetin as antiplatelet aggregation agents was evidenced by the impedance aggregometry assays. The effect observed was similar in WB and PRP, albeit the antiaggregant activity was superior in the latter. In both cases, the effect was more remarkable in AA-induced than in ADP-induced activated samples. Coumarin and esculetin showed an inhibition of 87% and 52%, respectively, in AA-induced samples at a concentration of 3 mM, whereas the inhibition was complete with highest concentrations.

With respect to coumarin, this compound was able to completely inhibit platelet agglutination in AA-induced PRP samples when a 1.5 mM solution was used. Regarding the effect of esculetin, the inhibition rate achieved a 16% at 3 mM, and was complete at 6.5 mM. The esculetin, in its heterosidic form as esculin, demonstrated a minimal decrease in platelet aggregation at 6.5 mM that was null at lower concentrations in WB and PRP, in either AA- or ADP-induced activated samples. Thus, our results revealed that the presence of the catechol group in esculetin favours the antiplatelet activity, while this activity is lost when the C-6 hydroxyl group is replaced by a sugar, such as in esculin.

The higher antiaggregant potency of aglycones vs. heterosides was confirmed in the COX-1 activity assay from the measurement of MDA when AA was employed as aggregant agent in PRP samples. Esculetin showed a 90% of inhibition of COX-1 activity at 6.5 mM, which was similar to the inhibition reached with the positive control indomethacin. Unlike esculetin, coumarin just showed a 60% of inhibition at the maximal concentration assayed and no significant effect was found for esculin.

However, the results obtained in the COX-1 inhibitory assay performed with the human pure recombinant enzyme were somewhat different. In this case, the heteroside esculin procured the greatest inhibitory effect on h-COX-1 since the enzyme activity decreased up to the 26%. Nevertheless, coumarin produced a 50% enzyme inhibition in a similar way to the inhibition showed by the MDA measurement assay. Esculetin, that had previously shown a 90% of COX-1 inhibition, returned a 57% of enzyme inhibition when tested with the pure enzyme. Regarding the indomethacin (positive control), the results with the pure enzyme seem to be more relevant than those related to the MDA production, since the enzyme inhibition reached a 95%.

Surprisingly, any of the coumarins investigated had an impact in TXB_2_ levels. Indomethacin, for its part, prevented TXB_2_ production in a 95%. Hence, it can be assumed that this molecule significantly inhibits COX-1 activity.

Coumarin and esculetin presented a dose-dependent effect on the platelet aggregation, the COX-1 activity and the pure enzyme. Higher concentrations of coumarins than those selected in this research work might exert a larger effect, but the low solubility of these compounds is a major limitation. Even though DMSO is an optimal solvent for these molecules, the use of higher amounts could affect the platelets integrity and is discouraged [30,31]. Previous experiments in our laboratory showed how a higher volume than 2 µL of DMSO could damage platelets contained in 1 mL of WB [32]. 

Polyphenols are naturally present in plants as O- and C-glycosides, while aglycones are not found in fresh plants but can occur after processing [33]. In general, the oral bioavailability of polyphenols is considerably limited [34,35]. As a consequence of enzyme hydrolysis, the heterosides lose the glycosidic moiety before reaching the bloodstream and can then pass through the cell membranes [36].

As previously indicated, our in vitro results show that in the specific case of coumarins, the aglycones present a greater antiplatelet effect than their heterosidic parent compounds.

Esculin did not show any activity in our in vitro experiments apart from those performed with the pure enzyme, which suggests the inability of this compound to access the platelet interior. This fact could be explained by the presence of a sugar ring in its chemical structure. Considering that glycoxidation favours the biological activity of coumarins [37], it could be hypothesized that esculin would present a similar activity to its aglycone in vivo. 

The smaller size of the coumarin and esculetin structures could ease their transport across the platelet membrane, and hence, produce a higher effect on the COX activity. However, our results support the feasibility of and need for future studies on the interaction of the coumarins with blood platelet membrane. Notwithstanding, these two molecules were not able to inhibit TXB_2_ production. Thereby, the results here presented point to a mechanism of action at a different level that would possibly involve TXA_2_ receptors.

## 4. Materials and Methods

### 4.1. Selected Compounds

The selected phenolic compounds coumarin, esculin and esculetin were purchased from Sigma-Aldrich (Sigma-Aldrich Chemical, Madrid, Spain) and dissolved in DMSO (dimethylsulphoxide) (Dismadel, Madrid, Spain) to a final concentration of 0.5 mM. This concentration is similar to the daily dose clinically used for the flavonoid diosmin (Daflon^®^ 500 mg) [28] and was established considering the structural similarity (presence of the benzo-α-pyrone core), the almost identical physicochemical properties and their comparable molecular weight between this drug and the compounds here investigated [38]. Further dilutions were performed to reach 0.75, 1.5, 3.0, 5.0, 6.5 mM for the different assays. To avoid altering the platelet configuration, the lowest volume of DMSO (Dismadel S.L., Madrid, Spain) that could ensure the dissolution of the compounds (2 µL) was added to the blood samples [32].

### 4.2. Study Cohort, Inclusion and Exclusion Criteria

Ten healthy volunteers (seven women and three men; aged 22.2 ± 1.2 [mean ± SD] years), none of which had undergone platelet function or complement activation treatment during the previous year, were recruited to furnish blood for every assay of this work. Participants were not included if they were smokers or showed any sign of kidney, lung, heart, or autoimmune disease, any chronic or acute infection, diabetes mellitus, a history of tumours, immunodeficiency or thrombocytopathy, hypercholesterolemia or were undergoing immunosuppressant, steroids, or nonsteroidal anti-inflammatory drug (NSAID) treatment. They were excluded if they had undergone any other treatment that could affect the platelet activity during the six months prior to the assay, anovulants included.

Written informed consent was signed by every participant. The study protocol was carried out in strict accordance with the guidelines of the 1975 Declaration of Helsinki, under approval of the Biomedical Ethics Committee of the University of Alcalá.

### 4.3. Peripheral Blood Extraction

Peripheral blood was collected by an antecubital puncture in sodium citrate-containing (3.8% wt/vol) Vacutainer^®^ tubes (Dismadel S.L., Madrid, Spain), discarding the first 2 mL. All extractions were performed at the Haematology Service of the Principe de Asturias Hospital, Alcalá de Henares (Madrid, Spain). Sodium citrate was selected as the anticoagulant instead of heparin, ethylenediaminetetraacetic acid (EDTA), or D-phenylalanyl-L-prolyl-L-arginine chloromethyl ketone (PPACK) given its lesser impact on the complement activation pathways [39].

### 4.4. Blood Samples Preparation

Platelet aggregation assays were performed in both, WB and platelet-rich plasma PRP [40]. The removal of blood cell components in PRP allows to better evaluate the effect of a specific compound on the platelets. Two different platelet activity inducers were employed: AA (Sigma-Aldrich Chemical, Madrid, Spain) 0.5 mM [41] and ADP (Sigma-Aldrich Chemical, Madrid, Spain) 5 µM [42]. The use of two different substances (AA and ADP) that promote platelet aggregation through different pathways, together with their administration on the different types of samples (WB and PRP), provides with a better understanding on the level of action of the different compounds under research. It was considered that the screened compounds could present different activities depending on the medium or the aggregation inducer. 

#### 4.4.1. WB Samples Preparation

Samples were kept at room temperature until use. They were homogenized in a plastic beaker and aliquots of 500 µL were distributed in aggregation Chronolog polyethylene cuvettes (Labmedics, Oxfordshire, UK) as soon as possible. After that, 500 µL of physiological saline solution (PSS) (Dismadel S.L., Madrid, Spain) were added to each cuvette. The diluted samples were incubated at 37 °C for 1 h with the selected compounds, or DMSO in the case of the control sample, in a thermostatic bath Unitronic 320 Selecta (Tecnylab, Madrid, Spain) to increase their solubility. Sample preparation and subsequent assays were performed in the first 3 h after blood extraction. 

#### 4.4.2. PRP Samples Preparation

Blood samples were subjected to centrifugation for 10 min at 1200 rpm twice in a centrifuge Jouan B-3.11 (Tecnylab, Madrid, Spain) and PRP was obtained by collecting the supernatant. Platelet counts were normalized to 200,000 platelets/µL PRP. Briefly, PRP was dissolved in the hematologic solvent Diluid 601 (Biolab Diagnostics, Barcelona, Spain) and counting was performed in a Neubauer cell chamber using a binocular microscope NIKON (Izasa S.L., Madrid, Spain). This method avoids some potential errors linked to automated cell counters, such as detection of bubbles as particles, the counting of cell components other than platelets or counting groups of platelets present in the sample. The refringence and morphology of platelets under the light microscope facilitated their unambiguous identification. Once the number of platelets in the original PRP samples was known, the calculated volume was transferred to the cuvettes and PSS was added until a final volume of 1 mL. Next, samples were incubated with the assayed compound or DMSO as a control for 5 min at 37 °C. 

### 4.5. Impedance Platelet Aggregometry Assay 

The procedure was carried out in a Chrono-Log 500 Lumi-Aggregometer (Labmedics, Oxfordshire, UK) connected to an Omnioscribe II data-logger, according to the manufacturer’s instructions. Only plastic material was used to be in contact with the samples. The experimental method is based on the measurement of the change in the electrical impedance (the resistance to the electric current) between two electrodes when platelet aggregation is induced by an agonist [40]. Thereby, the electrodes immersed in the WB or PRP samples continuously stirred at 1200 rpm become covered by a platelet monolayer. The impedance remains constant in the absence of an aggregation agent. On the contrary, the addition of an aggregant promotes the adhesion and agglutination of the platelets in the electrodes and produces an increase in the impedance that can be used as a measurement of the platelet aggregation.

### 4.6. MDA Quantification and COX Activity Assessment

Plasma MDA levels reflect COX activity and can be used as a qualitative test of platelet function or to quantify the effect of COX inhibitors. MDA is a product of the arachidonic acid metabolism in platelets that can be measured by spectrophotometric techniques. Absorbance is recorded at 532 nm to ensure that it is entirely due to the released MDA. The molar extinction coefficient of MDA at 532 nm is 1.56 × 10^5^ [29].

MDA analysis was performed in AA-induced activated PRP samples [31], since the MDA absorbance levels in the ADP-induced activated samples were much lower than those corresponding to total COX activity. 

Calibration curves were created by preparing a set of solutions with known concentrations of MDA and measured in a UV/VIS Philips PU 8700 spectrophotometer at 532 nm to later extrapolate the MDA levels in PRP. The curves were prepared in PSS and PRP without aggregant to establish the possible MDA release by nonactivated platelets. 

Curves ranged from 100–1000 nM to 1–10 µM and had regression coefficients near 1 (0.997 y 0.994, respectively) (Figure 6).

Test samples were processed following a similar procedure to the samples used to obtain the calibration curves. A volume of 375 µL of 40% trichloroacetic acid (ATA) (Sigma-Aldrich Chemical, Madrid, Spain) was added to propylene tubes containing the AA-induced activated PRP samples and gently mixed to mediate the protein precipitation. The tubes were covered to prevent oxidation. PSS was added up to a final volume of 2 mL, like in the samples used in the calibration curves. After centrifugation at 3500 rpm for 10 min, the supernatant was filtered through glass wool. One more centrifugation step was performed in similar conditions and then 0.12 M tiobarbituric acid (TBA) (Sigma-Aldrich Chemical, Spain) was added in a relation of 0.2 volumes per volume of acid supernatant.

Next, the capped tubes were heated in a water bath at 100 °C for 15 min. Once cooled down, the spectrophotometric measures were obtained at 532 nm.

MDA concentrations in control samples were considered as 100% of COX-1 activity and the concentrations in the test samples were expressed as the percentage of COX-1 activity. Indomethacin (Sigma-Aldrich Chemical, Madrid, Spain) was employed as positive control at the following concentrations: 0.00257, 0.006, 0.013, 0.019, 0.02 mM. These concentrations were set considering that the therapeutic concentration is 0.004 mg/mL [43]. The assayed compounds were dissolved in DMSO (2 µL) and added to the reaction medium at the concentrations previously mentioned: 0.75, 1.5, 3.0, 5.0, 6.5 mM.

### 4.7. Procedure for the COX-1 Inhibitory Assay

Human purified COX-1 enzyme with a purity of 95% was purchased from Vitro S.A. (Madrid, Spain). The enzyme was supplied in 10KU vials prepared in Tris-HCl 80 Mm and 1% Tween 20. The enzyme unit (EU) is defined as the amount of enzyme required to produce a change of 0.001 mn^−1^ in the optical density at a wavelength of 610 nm. According to the manufacturer, the vials were stored frozen and kept in the dark on ice during the assays. One hundred EU (4 µL) were added to the samples. AA was used as enzyme substrate to replicate the aggregation inducer employed in the impedance aggregometry and MDA quantification experiments.

The enzyme activity was determined by a chromogenic method based on the oxidation of N,N,N′,N′-tetramethyl-p-phenylenediamine (TMPD) [44]. Among the substances produced during AA-induced platelet activation, the prostaglandin G_2_ (PGG_2_) is quickly reduced to PGH_2_ by the platelet enzyme COX-1. Because of this reduction, the TMPD is oxidized in a directly proportional amount to the enzyme activity.

The experimental studies were carried out in solutions containing the pure enzyme incubated with the test compounds and AA as the enzyme substrate. Absorbance produced by TMPD was measured at 610 nm in a Biotek ELx800 Absorbance Microplate Reader (Izasa Scientific, Madrid, Spain).

The screened compounds were prepared in DMSO at the concentrations abovementioned for the previous assays. Indomethacin, a selective COX-1 inhibitor, was used as control drug at the therapeutic concentrations of 0.001, 0.0025, 0.005, 0.0075, 0.01mg/mL [43].

### 4.8. Enzyme Immunoassay for the Quantitative Determination of TXB_2_

TXA_2_ is produced from AA oxidation and physiologically active. However, it is rapidly hydrolysed (average life of 30 s) to form TXB_2_, a stable and biologically inactive metabolite [45]. TXB_2_ concentration, as measured by immunoassay, is maximal at 20–30 min and declines thereafter [46], being considered as a measurement of the TXA_2_ levels. For this reason, the samples were incubated in WB and calcium ionophore A23187 (CI) (Sigma Aldrich, Madrid, Spain) was added at a concentration of 25 mM [32] to trigger platelet activation that produces TXB_2_. In this way, it can be considered as an indirect measure of the COX-1 activity inhibition [46].

TXB_2_ levels were determined by a specific enzyme immunoassay kit (TXB_2_ Biotrak Enzymeimmunoassay System, Amersham Biosciences, Little Chalfont, Buckinghamshire, UK) according to the manufacturer´s protocol. This kit possesses a high sensitivity (0.2pg) and the standard curve ranges from 0.5 to 64 pg. The absorbance values were obtained in a Biotek ELx800 Absorbance Microplate Reader (Izasa Scientific, Madrid, Spain) coupled to an automatic microplate washer Biotek ELx50 (Izasa Scientific, Madrid, Spain).

Similarly to prior assays, indomethacin was used as control drug because of its ability to inhibit COX [43].

To perform the procedure, the total volume of WB from donors was distributed in equal aliquots of 1 mL and incubated in a thermostatic bath (Unitronic 320 Selecta) (Izasa Scientific, Madrid, Spain) at 37 °C for 1 h with 2 µL of the test solutions or DMSO as control. After that time, 2 µL of CI 25 mM were added and incubation maintained for a further 30 min cycle. The reaction was terminated by introducing the samples on dry ice. Then, the samples were centrifuged (centrifuge Jouan 3.11) (Tecnylab, Madrid, Spain) at 4000 rpm for 10 min and the supernatant collected and subjected to the enzyme-linked immunosorbent technique.

### 4.9. Statistical Analysis

All results are expressed as the mean ± standard deviation (SD) of values obtained in each experiment. Since most variables did not fulfil the normality hypothesis, the Wilcoxon test was used to analyse the variance of paired groups. The level of significance was set at *p* < 0.05. Statistical analysis was performed using SPSS-27.0 software (SPSS-IBM, Armonk, NY, USA).

## Figures and Tables

**Figure 1 molecules-26-03036-f001:**
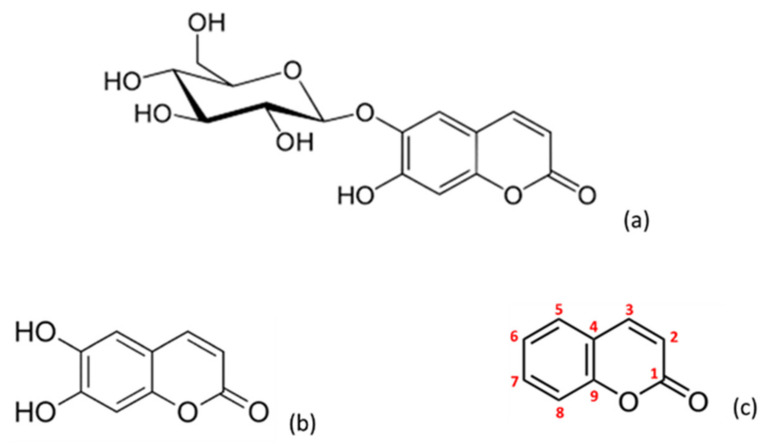
Chemical structure of the different coumarins assayed: esculin (**a**), esculetin (**b**) and coumarin (**c**).

**Figure 2 molecules-26-03036-f002:**
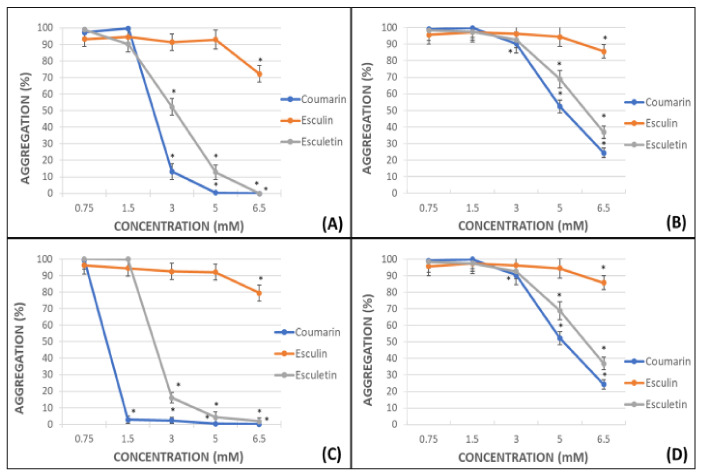
Graphical representation of the percentage of platelet aggregation for coumarin, esculin and esculetin. Panels (**A**) (WB samples) and (**C**) (PRP samples) shows results in AA-induced platelet aggregation. Panels (**B**) (WB samples) and (**D**) (PRP samples) shows results in ADP-induced platelet aggregation. Results are expressed as the mean and standard deviation for 10 donors. Error bars represent the standard deviation. * *p* < 0.05: statistically significant differences in platelet aggregation between samples with and without the tested phenolic compound.

**Figure 3 molecules-26-03036-f003:**
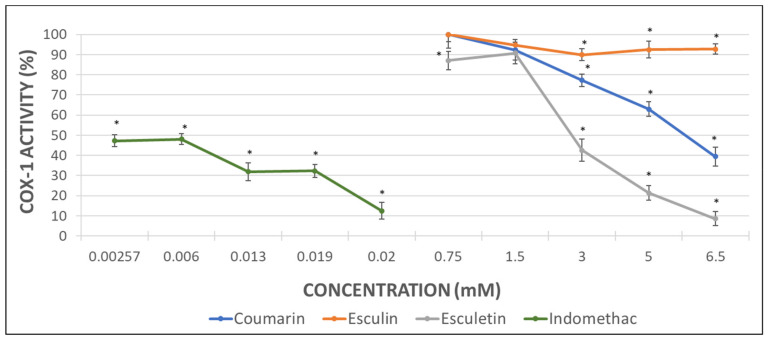
Graphical representation of the percentage of COX activity in AA-induced activated PRP samples after addition of increasing concentrations of indomethacin, as positive control, and assayed coumarins. Results are expressed as the mean and standard deviation for 10 donors. Error bars represent the standard deviation. * *p* < 0.05: significant differences regarding COX activity with and without the examined substances.

**Figure 4 molecules-26-03036-f004:**
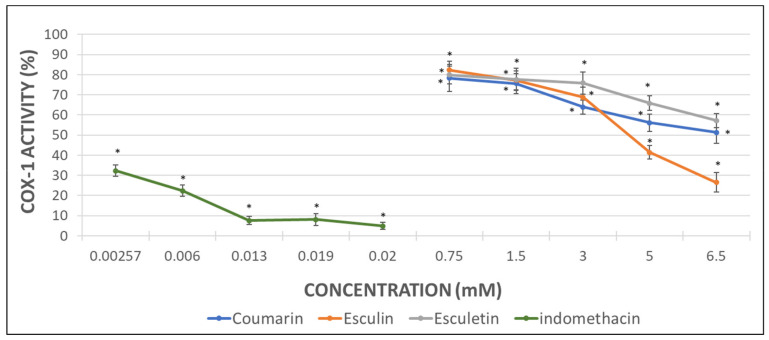
Graphical representation of the percentage of h-COX-1 activity after indomethacin and tested coumarins addition in AA-induced activated samples. Results are expressed as the mean and standard deviation for 10 donors. Error bars represent the standard deviation. * *p* < 0.05: significant differences on h-COX-1 activity with and without the examined substances.

**Figure 5 molecules-26-03036-f005:**
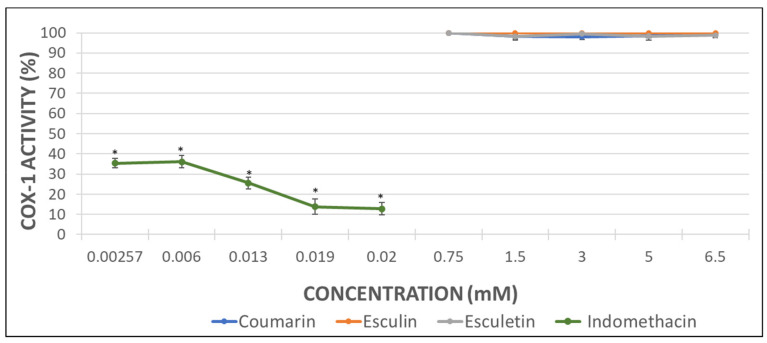
Graphical representation of the percentage of TXB_2_ production at different concentrations of indomethacin and coumarins after CI-induced aggregation. Results are expressed as the mean and standard deviation for 10 donors. Error bars represent the standard deviation. * *p* < 0.05: significant differences between the basal TXB_2_ production with and without the examined substances.

**Figure 6 molecules-26-03036-f006:**
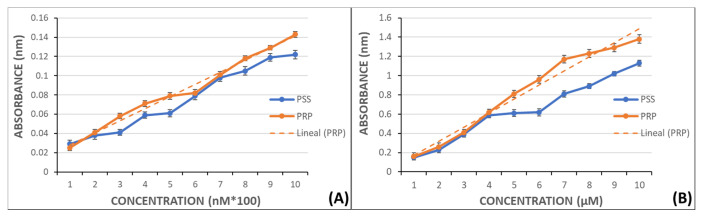
Graphical representation of MDA absorbance in PSS y PRP. Panel (**A**): in the range from 100 nM to 1000 nM and linear fitting in PRP samples (r = 0.997). Panel (**B**): in the range from 1 µM to 10 µM and linear fitting in PRP samples (r = 0.994).

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
