# Peer review of "Antiplatelet Activity of Coumarins: In Vitro Assays on COX-1"

_molecules, 2021, doi:10.3390/molecules26103036_

Round 1

Reviewer 1 Report

In this article by Zaragozá et al., Antiplatelet aggregation activities of coumarin, esculetin and esculin were determined in vitro in human whole blood and platelet-rich plasma.

Comments:

1. The esculin and esculetin compounds were selected based on any previous activity?
2. What is 2mcl? correct
3. What is mcM? or is uM, correct
4. The methodology should be ordered. For example, first the collection of the sample (PRP) and then the aggregation test. Because it is described twice 4.4. Impedance platelet aggregometry and 4.4.3. Impedance platelet aggregometry assay?
5. Improve the quality of the figures
6. The article has too many figures, they should be grouped by technique, for example platelet aggregation in a figure, separating with letters (figure 1 A, 1B, 1C ...)

Reviewer 2 Report

The work entitled “Antiplatelet activity of coumarins. In vitro assays on COX-1” aimed to provide insight into the biochemical mechanisms of the coumarins-blood platelets interactions. However, the manuscript content only partly reflects the study hypothesis. The examined coumarins were used at very high concentrations, i.e. 0.75 – 6 mM, that are physiologically unachievable.

Moreover, blood platelets are very prone to cytotoxic action of different substances, thus, the use of milimolar concentrations of coumarins is associated with a significant risk of cytotoxic action. The anti-platelet effect of the tested substances may be partly a result of their cytotoxic action towards these cells, but the manuscript does not contain any data on cytotoxicity tests.

Due to a poor methodology, a major revision is needed. The authors provided simple tests without any mechanistic insight into the observed effects. The study should be extended by at least one or two more analytical methods, in order to enhance the quality of this work. I suggest some additional experiments devoted to the coumarins-COX-1 activity or coumarins - blood platelets interactions. It could be either in vitro or in silico study.

The presentation of results is confusing. Data for the examined substances and the reference drug (i.e. indomethacin) should be presented in the same figures.

The IC50 parameter should be established for both the examined substances and reference drug.

All the remaining, specific concerns and suggestions have been included as my comments into the manuscript draft.

Round 2

Reviewer 1 Report

The article has been considerably improved.

Author Response

Thank you very much

Reviewer 2 Report

The revised version of a manuscript “Antiplatelet activity of coumarins. In vitro assays on COX-1” has been partly improved, however, it still needs several corrections.

One of the most important concerns is the used concentration range (0.75 – 6 mM). The mentioned concentrations are physiologically unachievable, and therefore, I asked the authors for justification of these doses. The authors still have not provided any data on cytotoxicity tests of the examined substances. Since the anti-platelet effect of the tested substances may be partly a result of their cytotoxic action towards these cells, these data are critical for reliable evaluation of the anti-platelet effects of the examined coumarins. The literature provides numerous reports confirming toxicity of coumarins to different cell types, and blood platelets are extremely prone to cytotoxic action of exogenous substances. For example, a significant cytotoxic effect of coumarins in cell lines can be attained at concentrations lower than 100 micromoles/mL (Esra Küpeli Akkol et al., Cancers 2020; doi:10.3390/cancers12071959). Dear Authors, please, include the flow cytometry results (accordingly to your kind response) to the manuscript to prove the blood-platelet safety of the examined compounds.

Other concerns:

The figure 2 legend: Please correct the following sentences ”Panels A (WB samples) and C (PRP samples) shows results in AA-induced. Panels B 104 (WB samples) and D (PRP samples) shows results in ADP-induced.” The information what “induced” should be given.

Also in the text below this figure (lines 109-110), there is no information what was induced.

Figure 3: Within this figure, the name of indomethacin has been cut. Please, replace “drugs” in the last sentence into “the examined substances” in the figure legend.

Figure 5: Please, replace “drugs” in the last sentence into “the examined substances” in the figure legend.
